# An Inflamed and Infected Reconstructed Human Epidermis to Study Atopic Dermatitis and Skin Care Ingredients

**DOI:** 10.3390/ijms232112880

**Published:** 2022-10-25

**Authors:** Sébastien Cadau, Manon Gault, Nicolas Berthelemy, Chiung-Yueh Hsu, Louis Danoux, Nicolas Pelletier, Dominique Goudounèche, Carole Pons, Corinne Leprince, Valérie André-Frei, Michel Simon, Sabine Pain

**Affiliations:** 1BASF Beauty Care Solutions France, 32 Rue Saint Jean de Dieu, 69007 Lyon, France; 2Centre de Microscopie Electronique Appliquée à la Biologie, Paul Sabatier University, 133, Route de Narbonne, 31062 Toulouse, France; 3Toulouse Institute for Infectious and Inflammatory Diseases (Infinity), CNRS UMR5051and Inserm UMR1291, CHU Purpan BP 3028, CEDEX 3, 31024 Toulouse, France

**Keywords:** atopic dermatitis, inflammation, microbiota, epidermal barrier, lamellar body, tissue engineering

## Abstract

Atopic dermatitis (AD), the most common inflammatory skin disorder, is a multifactorial disease characterized by a genetic predisposition, epidermal barrier disruption, a strong T helper (Th) type 2 immune reaction to environmental antigens and an altered cutaneous microbiome. Microbial dysbiosis characterized by the prevalence of *Staphylococcus aureus* (*S. aureus*) has been shown to exacerbate AD. In recent years, in vitro models of AD have been developed, but none of them reproduce all of the pathophysiological features. To better mimic AD, we developed reconstructed human epidermis (RHE) exposed to a Th2 pro-inflammatory cytokine cocktail and *S. aureus*. This model well reproduced some of the vicious loops involved in AD, with alterations at the physical, microbial and immune levels. Our results strongly suggest that *S. aureus* acquired a higher virulence potential when the epidermis was challenged with inflammatory cytokines, thus later contributing to the chronic inflammatory status. Furthermore, a topical application of a *Castanea sativa* extract was shown to prevent the apparition of the AD-like phenotype. It increased filaggrin, claudin-1 and loricrin expressions and controlled *S. aureus* by impairing its biofilm formation, enzymatic activities and inflammatory potential.

## 1. Introduction

Atopic dermatitis (AD) is a chronic skin inflammatory disease [1], highly affecting children. The disease generally improves or disappears with age, but when it persists into adulthood, it is usually considered severe. Interestingly, most cases have been reported in urban areas of developed countries, where excessive hygiene, psychological factors and pollution seem to be key pathological factors. The severity of the disease is ranked with an Eczema Area and Severity Index (EASI) score based on local cutaneous manifestations [2].

In healthy epidermis, the barrier function has several purposes: to protect against ultra-violet radiation; to maintain good hydration of the upper cell layers and, at the same time, limit body fluid and water loss; to provide an immune defense system against microbial infection; and, finally, to physically and chemically control what goes inside the skin [3]. To ensure these functions, the epidermis is renewed every 21–30 days from the bottom to the top. The proliferation of keratinocytes is restricted to the basal layer, when cell differentiation is gradually being set up in the upper cell layers. In the upper spinous keratinocytes, lipid molecules start to be synthetized and packaged into lamellar bodies. In the granular layer, keratinocytes flatten, tight junction complexes are formed to control small molecule diffusion between cells [4], and lamellar bodies fuse with the plasma membrane to release their content (lipids: ceramides, cholesterol, free fatty acids; proteases; corneodesmosin; anti-microbial peptides, etc.) in the extracellular spaces [5,6]. Finally, in the *stratum corneum*, the outermost layer of the epidermis, anucleated corneocytes are stacked and embedded in a lipid-enriched extracellular matrix.

Various studies have demonstrated that AD is characterized by a default in the epidermal barrier [7,8]. Two non-exclusive hypotheses face off to explain the pathology:The inside-out signal, where chronic inflammation, driven by T helper (Th)2 cytokines, including interleukin (IL)-4 and IL-13, secondarily alters keratinocyte differentiation and reduces the expression of several epidermal barrier proteins [9];The outside-in signal, where epidermal barrier disruption allows for the penetration of allergens and microbes and triggers immunological imbalance [10,11,12].

In addition, IL-31 is described as being responsible for pruritus, but it is not directly linked to inflammation and pain [13]. In favor of the outside-in hypothesis, mutations in several epidermal barrier genes have been strongly associated with AD susceptibility, including loss-of-function mutations of the filaggrin gene [14,15,16] and mutations in the non-coding part of the claudin-1 gene [12]. Filaggrin is a major player in the epidermal barrier [17]. Its precursor, profilaggrin, the main constituent of keratohyalin granules, is proteolyzed into multiple filaggrin monomers that bind and aggregate intermediate filaments. Cellular compaction and extensive protein crosslinking occur leading to the formation of a dense intracellular keratin matrix, where cornified envelope proteins are cross-linked. Filaggrin expression is reduced in both the apparently normal and lesional skin of patients with AD, irrespective of its gene mutations. Claudin-1, with claudin-4, is the major constituent of tight junctions in the epidermis. Its absence leads to embryonic death due to water loss and an abnormal skin phenotype [18]. Its expression is also decreased in AD [12]. Loricrin, initially expressed in the granular layer, represents 80% of the total protein mass of cornified envelopes. In patients with AD, its expression appears to be downregulated by Th2 cytokines [11]. Abnormal lamellar body secretion has also been described in AD [19,20], leading to alterations in the content of *stratum corneum* extracellular lipids and, therefore, in the permeability properties of this layer [21].

As a consequence of both epidermal barrier impairments and changes in the (bio)chemical properties of the *stratum corneum*, the homeostatic microbiota balance of the skin evolves and does not prevent the growth of pathogens, such as *Staphylococcus aureus* (*S. aureus*) [22,23]. Thus, secreted virulence factors contribute to the pathogenesis of AD, activating inflammation and the immune response. Moreover, *S. aureus* express superantigens, which are allergens and bind antigen-presenting cells and T cell receptors [24].

Over the years, various models of AD have been developed [25]. Some have used cells from patients to understand the respective roles of fibroblasts and keratinocytes in the disease [26]. Others have used Th2 cytokines, such as IL-4 and IL-13, to reproduce the AD phenotype [27,28]. Genomic approaches have been investigated, including the knockout [29] or knockdown of filaggrin [30,31]. Finally, the impact of bacterial components on a compromised skin barrier has been shown by adding *S. aureus* onto reconstructed human epidermis (RHE) [32].

In the present study, we developed a new 3D model of RHE reproducing AD, adding different concentrations of both Th2 cytokines and *S. aureus*. We demonstrated that *S. aureus* adhesion and proliferation were enhanced in the presence of the cytokines, with the bacteria inducing a strong inflammation response as evidenced by an increased IL-8 release. We also used the model to evaluate the protective effect of an active ingredient from the leaves of *Castanea sativa*. This active ingredient was preselected from a library of thousands of potential ingredients by performing a preliminary anti-inflammatory property evaluation (quantification of IL-6 and IL-8 by ELISA) on monolayer cultures of normal human keratinocytes stressed by poly I:C, TNF alpha and interferon gamma. Among the anti-inflammatory hits, *Castanea sativa* extract was then selected for its capacity to inhibit IL-6 and -8 production in keratinocytes challenged by *S. aureus* ATCC35556, to inhibit their lipase activity and, in parallel, to stimulate filaggrin on differentiated keratinocytes.

## 2. Results

### 2.1. AD-like RHEs Treated with S. aureus and/or Cytokines

We generated two in vitro models of AD using RHEs treated with either a cocktail of inflammatory cytokines in a culture medium or both the cocktail and topically applied *S. aureus*. We used TNFα at 5 ng/mL and the Th2-related cytokines IL-4, IL-13 and IL-31 at two concentrations, 5 ng/mL each (C1) and 20 ng/mL each (C2), in order to attempt to reproduce a moderate-to-mild phenotype.

The morphology of the obtained epidermis was analyzed by hematoxylin–eosin staining (Figure 1a,e,i,m,q). The non-treated RHEs showed a well-formed orthokeratotic *stratum corneum*, a good organization of the *stratum spinosum* and *granulosum* and a nice basal layer with well-shaped pavement cells (Figure 1a). In all treated RHE conditions, complete epidermal maturation occurred, with the presence of a normal *stratum corneum*. However, after the addition of the highest concentration of the inflammatory cocktail, we observed the disorganization of all keratinocyte layers, the appearance of spongiosis and a strong reduction in keratohyalin granules (Figure 1i). A more drastic effect (the presence of apoptotic cells, with no evidence of whether a basal layer existed) was observed when *S. aureus* was used in the presence of 5 ng/mL of cytokines, a concentration that appeared to not have any major effects by itself (Figure 1e,m).

To determine whether AD-related alterations in the expression of keratinocyte differentiation markers can be obtained in vitro, sections of the RHEs were immunostained with anti-filaggrin (Figure 1b,f,j,n,r), anti-loricrin (Figure 1c,g,k,o,s) and anti-claudin-1 (Figure 1d,h,l,p,t) antibodies. Filaggrin expression was slightly impaired at the lowest concentration of cytokines, with a possible additive effect when combined with *S. aureus* (Figure 1f,n), and it appeared to massively decrease at the highest concentration of cytokines (Figure 1j). Very similar effects were observed on the expressions of loricrin and claudin-1. In addition, the claudin-1 location at the cell periphery was lost, especially in the upper part of the epidermis, at the highest concentration of cytokines. The expression of the three proteins was then quantified using Western blotting experiments (Figure 2). When the RHEs were treated with the lowest inflammatory cocktail concentration associated with *S. aureus*, the expressions of filaggrin, claudin-1 and loricrin were reduced compared to those in the control condition by 49%, 14% and 73%, respectively.

### 2.2. Treatment of RHE with Cytokines Alters Its Morphology at the Ultrastructural Level

Transmission electronic microscopy observations were carried out to look closely at the ultrastructure of the RHE (Figure 3). The non-treated RHEs showed the classical ultrastructural morphology of interfollicular epidermis, with corneocytes filled with an electron-dense matrix and granular keratinocytes containing large and stellate keratohyalin granules and lamellar bodies (Figure 3a,d). In the RHEs treated with the highest inflammatory cocktail dose (C2), several observations were made: the number and size of keratohyalin granules were drastically reduced, the lamellar bodies were scarce, and an atypical accumulation of vesicles was found (Figure 3b,e).

### 2.3. Cytokine Treatment Promotes S. aureus Adhesion and Growth

As shown above, the application of *S. aureus* on the RHEs exacerbated the defects induced by the lowest concentration of cytokines at the morphological level and at the expression level of key epidermal proteins (Figure 2). To determine whether cytokines impact the colonization of the epidermal surface by *S. aureus*, an equal amount of this pathogen was topically inoculated on the non-treated and Th2-cytokine (5 ng/mL)-treated RHEs. Its adhesion and proliferation were measured (Figure 4). One hour after the application, only 5% of the initial inoculum remained on the surface of the non-treated RHEs. This quantity increased 6.6 times when *S. aureus* was seeded on the inflamed RHEs, suggesting that the cytokines enabled better adhesion. Twenty-four hours later, although roughly no bacteria were detected on the surface of the non-treated RHEs, the bacteria on the surface of the treated RHEs showed exponential growth, with 115 times more *S. aureus* than in the inoculum, demonstrating that the cytokine cocktail favored *S. aureus* growth.

### 2.4. Topically Applied S. aureus Increases the Secretion of IL-8

The release of IL-8, an innate inflammatory cytokine, was quantified in the culture medium (Figure 5). The lower concentration (5 ng/mL, C1) of the cytokine cocktail did not modify the concentration of IL-8, whereas the higher concentration (20 ng/mL, C2) induced a 6.5 times increase. When *S. aureus* was topically applied on the RHEs treated with 5 ng/mL of cytokines, a 3.1 times increase was noted.

### 2.5. A Castanea Sativa Extract Reverses the AD-like Phenotype of Th2-Cytokine- and S. aureus-Treated RHEs

To demonstrate the capability of the newly developed model of AD, which is based on cytokine- and *S. aureus*-treated RHEs, we used it to test the potential protective properties of an extract obtained from *Castanea sativa* leaves. When the extract was added to the culture medium at a concentration of 0.04% during the last 7 days of culture, a clear improvement in the phenotype was observed at the morphological level after hematoxylin–eosin staining (Figure 1m vs. Figure 1q) and the transmission electron microscopy analysis (Figure 3b vs. Figure 3c). The induced downregulation of filaggrin, loricrin and claudin-1 expressions was reversed, as shown using indirect immunofluorescence (Figure 1r–t) and the Western blotting (Figure 2) analysis. The secretion of IL-8 was also reduced by 30% (Figure 5).

### 2.6. The Castanea Ssativa Extract Reduced S. aureus Virulence

In order to confirm the protective potential of the *Castanea sativa* extract, we addressed its effects on the virulence properties of *S. aureus* using in vitro models reflecting biofilm formation, the secretion of cofactors and exoenzymes and the production of inflammatory cytokines by monolayer cultures of human keratinocytes and macrophages.

Compared to the control cultures of *S. aureus*, the formation of biofilm, as quantified after crystal violet staining, decreased in a dose-dependent manner by 78% and 96% when treated with 0.5% and 1.5%, respectively, of the *Castanea sativa* extract (Figure 6a). This result was confirmed with the monitoring of the electric impedance in the *S. aureus* culture over time (Figure 6b). In the control conditions, after an initial phase of 4 h, during which impedance decreased, a drastic increase in biofilm formation was observed, with a maximum reached 19 h post-seeding (cell index of 0.22). Although an identical initial phase occurred, when the *S aureus* cultures were treated with the *Castanea sativa* extracts, a dose-dependent reduction in biofilm formation was observed (cell indexes of 0.03 and ∼0.0). This inhibition became significant after 8 h of culture and lasted up to 24 h.

We then measured the effect of various concentrations (from 5.5% (*v*/*v*) to 0.015%) of the *Castanea sativa* extract on the enzymatic activities (lipase, hyaluronidase and plasminogen activator) released by *S. aureus*. The plant extract significantly inhibited the three enzymatic activities with various efficacies. We observed a high inhibition of lipase activity (IC50 = 0.33%) and plasminogen activation (IC50 = 0.035%) and a moderate inhibition of hyaluronidase activity (20% inhibition when the extract was present at 1.5%).

Finally, we tested whether the plant extract was able to reduce the amount of inflammatory cytokines released by the *S. aureus*-infected human keratinocytes and macrophages. A strong increase in IL-6 and IL-8 production by keratinocytes (HaCaT human cell line) was induced by the bacteria (Figure 6c). This increase was dose-dependently reduced with the extract, with the highest dose (0.03%) inducing an almost complete reversion of the IL-6 level to the control level and a 76% reduction in the IL-8 level (Figure 6c). *S. aureus* infection induced a 60% increase in the level of IL-8 released by macrophages (Figure 6d). This effect was partially but significantly and dose-dependently inhibited by the *Castanea sativa* extract, with reductions of 32 and 55% when the extract was used at 0.01% and 0.03%, respectively (Figure 6d). The measure of cell viability by the MTT test demonstrated that the Castanea sativa extract was nontoxic at the tested concentrations.

## 3. Discussion

AD pathogenesis is mainly driven by an impaired epidermal barrier that allows an increased penetration of allergens and irritants into the skin and a favorable ground for opportunistic bacteria proliferation. These phenomena are responsible for strong inflammation, skin redness, dryness and pruritus. Skin barrier defects are characterized by an increase in trans-epidermal water loss and the impaired expression of keratinocyte terminal differentiation markers, including filaggrin; loricrin; and the components of tight junctions, such as claudin-1.

To develop an in vitro tridimensional AD model, we treated RHEs with a cocktail of inflammatory cytokines, including TNFα and the Th2-related cytokines IL-4, IL-13 and IL-31. In addition, with the aim to mimic mild and severe AD, the cocktail was used at two different concentrations. These cytokines are known to induce inflammation and pruritus, but they are also known to have negative effects on keratinocyte differentiation [33,34,35]. We showed that, the more concentrated the inflammatory cocktail, the greater the decrease in the expression of the epidermal differentiation markers. Ultrastructural analyses of RHE treated with the highest concentration of the inflammatory cocktail showed modification of the *stratum granulosum*, with a decreased number and size of keratohyalin granules, the presence of atypical vesicles and a spongiosis aspect, hallmarks of the epidermis of patients with AD. Thus, we can speculate that the occurrence of impaired epidermal proteins and the lipid “mortar” of patients with AD leading to skin dryness could be one of the consequences of TNFα and Th2 cytokine actions [28,36,37].

Furthermore, the microbiota influence has to be taken into account in AD in vitro models. Indeed, *S. aureus* is considered a major cause of exacerbation of the pathology, and a strong correlation between AD severity and the density of *S. aureus* on lesional and non-lesional areas has been reported [38]. Biofilm formation is a dominant strategy used by *S. aureus* to survive and infect human skin [39]. Its development consists of an initial attachment phase and then a maturation phase combining bacteria growth and biofilm stabilization through the release of an extracellular matrix. Finally, the detachment/dispersal phase enables the dissemination of bacteria and, thus, the infection of new sites [40]. The severity of skin atopic lesions is directly correlated to *S. aureus* biofilms that cause sweat duct occlusion and skin irritation [39]. A vicious circle between inflammation and *S. aureus* growth and infection exists in the pathology.

Using our AD-like RHE model, we showed that treatment with a low concentration of the inflammatory cocktail and the topical application of the bacteria had cumulative negative effects on RHE morphology and on the expression of keratinocyte proteins. Moreover, we showed that a mild inflammation of RHE was beneficial for S. aureus adhesion to keratinocytes, as well as its growth on the surface of the epidermis. This is in accordance with previous reports showing that Th2 cytokines promote *S. aureus* skin colonization through various processes, including reduced anti-microbial peptide synthesis [41]. This step is crucial in patients with AD because bacteria virulence factors, including toxins and proteases, consequently induce deleterious effects on the epidermis (keratinocyte lysis and proinflammatory cytokine production), leading to enhanced skin inflammation [42]. The virulence of *S. aureus* is associated with a wide range of secreted factors, such as lipase [43], staphylokinase [44] and hyaluronidase [45]. *S. aureus* lipase promotes skin colonization through the release of irritant free fatty acids from sebum triglycerides, thus compromising skin barrier function [46]. Staphylokinase is a plasminogen activator, thus facilitating *S. aureus* local spread, reducing bacterial clearance and increasing skin tissue damage [47]. Secreted hyaluronidase is known as a spreading factor for bacteria, and it can increase the skin damage caused by *S. aureus* [45]. Published data also showed that *S. aureus* superantigens induce cytokine production [48]. *S. aureus* induces inflammatory cytokine release from keratinocytes by binding to Toll-like receptors [49]. Moreover, as a coactivator of the disease, macrophages accumulate and persist as pro-inflammatory resident cells in atopic skin [50]. The activation of macrophages by staphylococcal exotoxins induces the release of inflammatory cytokines, including IL-8, the level of which is increased in AD [51]. We showed in our model that *S. aureus* increased the release of IL-8, confirming the relevance of this model to AD pathophysiological studies and to active ingredient screening.

Finally, we used our model to evaluate the efficacy of an extract of *Castanea sativa* leaves. This plant extract was shown to prevent the defects in the epidermal barrier induced by *S. aureus* and low levels of cytokines through the maintenance of a better epidermal organization and differentiation. Moreover, in vitro methods further highlighted the potential of the plant extract to reduce *S. aureus* adhesion, biofilm formation and the enzymatic activities secreted by *S. aureus*, contributing to preserve a healthy microbiome. Complementary results showed that cytokine production by keratinocytes (IL-6 and IL-8) and macrophages (IL-8) infected by the bacteria was decreased when treated with the *Castanea sativa* extract. With the achievement of this key step, we evaluated the efficacy of the plant extract at 2% on a cohort of 22 volunteers with mild-to-moderate AD. The significant improvement in the barrier function (trans-epidermal water loss values decreased by 21% and 27% versus baseline at 1 and 2 months, respectively) and EASI score (decreased by 39% and 49%) confirmed the relevance of our model to evaluate ingredients that can be used to improve atopic-prone skin. The activity of the *Castanea sativa* extract may be explained by its flavonoid profile, particularly flavonoid glucosides, and among them, the major compounds miquelianin (Quercetin-3-O-glucuronide) and astragalin (Kampferol 3-O- glucuronide), as these two compounds are known as NF kB pathway modulators [52,53,54,55].

## 4. Materials and Methods

### 4.1. Castanea sativa Extract

CaVa is an aqueous extract of the leaves of the *Castanea sativa* tree. The organic certified leaves are harvested as a byproduct of chestnut production. Raw materials were grinded, followed by aqueous extraction and filtration. The extraction process was optimized to standardize flavonol glycosides: the amount of astragalin, miquelianin and equivalents.

### 4.2. RHE Production and Treatments with Cytokines, Castanea sativa Extract and Bacteria

Tridimensional RHEs were produced on a polycarbonate insert, as previously described [56], with primary human keratinocytes from abdominal skin (Centre de Ressources Biologiques, Edouard Herriot Hospital, Lyon, France). Next, 3 × 10^5^ keratinocytes/cm^2^ were seeded on the membrane and grown in Epilife medium containing 1.5 mM CaCl_2_, HKGS and antibiotics (Thermofisher, Illkirch-Graffenstaden, France) by immersion for 2 days at 37 °C with 5% CO2. The inserts were then elevated at the air–liquid interface, and the culture was continued for 12 days with supplementation with 50 µg/mL ascorbic acid and 10 ng/mL KGF (Merck, Molsheim, France). The fully supplemented Epilife medium was changed every 2 days. When necessary, RHEs were treated, from day 5, with an inflammatory cocktail composed of 5 ng/mL of TNFα and either 5 ng/mL (C1) or 20 ng/mL (C2) of each of the following cytokines: IL-4, IL-13 and IL-31. The RHEs were also treated with a *Castanea sativa* leaf extract (Castaline™, Basf, Pulnoy, France) at 0.04% from day 5 to 11 in a systemic way.

The RHEs treated with C1/C1+ CaVa were challenged in parallel by *S aureus*. For these experiments with a *S. aureus* strain obtained from a lesion in the thigh of a 44-year-old patient with AD (CIRI, Lyon, France), antibiotic treatment was stopped at day 10, and 130 µL of PBS containing 5 × 10^2^ bacteria was seeded on top of the RHEs at day 11. After 1 h of contact, bacteria were washed twice with PBS. The RHEs were harvested immediately or 24 h later. Bacteria were counted using the tryptic soy agar plate method (Biomérieux, Marcy-l’Étoile, France). Each condition was carried out in triplicate.

### 4.3. Histological and Immunohistological Analyses of RHE Sections

RHEs were formaldehyde-fixed and paraffin–embedded, and hematoxylin–eosin staining was performed on 5 µm-thick sections using a standard histochemistry technique. Immunofluorescence staining for the detection of filaggrin, claudin-1 and loricrin were carried out on 5 µm-thick sections using anti-filaggrin (Abcam #ab218395, Cambridge, UK), anti-claudin-1 (Invitrogen #71-7800, Villebon-sur-Yvette, France) and anti-loricrin (Abcam ab198994, UK) primary antibodies, followed by secondary antibodies conjugated to either Alexa 488 or Alexa 555. After several washes and counterstaining with 4′,6-diamidino-2-phenylindole (DAPI Vector), sections were mounted with an anti-fading medium. Observations were realized using bright-field (DMRBE, Leica, Wetzlar, Germany) or confocal (TCS SP2, Leica) microscopy.

### 4.4. Transmission Electron Microscopy

As described by Reynier et al., the RHE tissues were fixed with 2.5% glutaraldehyde and 2% paraformaldehyde in 0.1 M cacodylate buffer, pH 7.2, for 24 h at 4 °C, and post-fixed at 4 °C with 1% OsO_4_ and 1.5% K_3_Fe(CN)_6_ in the same buffer [57]. The tissues were treated for one hour with 1% aqueous uranyl acetate, dehydrated in a graded acetone series and embedded in EMbed 812 resin (EMS, Hatfield, PA, USA). After 48 h of polymerization at 60 °C, ultrathin sections (80 nm) were mounted on 75 mesh Formvar/carbon-coated copper grids. The sections were stained with uranyl acetate and lead citrate. The grids were examined with a TEM (Jeol JEM-1400, Peabody, MA, USA) at 80 kV. Images were acquired using a Gatan Orius digital camera (Pleasanton, CA, USA).

### 4.5. Western Blotting Analysis

After treatments, RHEs were lysed with specific lysis buffer (T-PER, Thermo Scientific, #78510, Waltham, MA, USA) to evaluate filaggrin, loricrin and claudin-1 expressions. Protein concentration was determined by a BCA assay, and then lysates were kept frozen at −80 °C until use. All the samples were adjusted to the same protein concentration, and an equal quantity of proteins was loaded in each capillary. Target proteins were identified by a capillary electrophoresis-based protein analysis system (Sally Sue; ProteinSimple, San Jose, CA, USA) using primary antibodies (anti-filaggrin, clone AKH1, Santa Cruz Biotechnologies #sc66192, Dallas, TX, USA; anti-loricrin, Abcam #ab176322; anti-claudin-1, Life technologies #71-7800, Carlsbad, CA, USA), and they were immunoprobed using a horseradish peroxidase-conjugated secondary antibody and chemiluminescent substrate. The capillaries containing a proprietary UV-activated chemical-linked coating were obtained from ProteinSimple. All samples and reagents were prepared according to the recommended manufacturer’s instructions. The resulting chemiluminescent signal was detected and quantified using Compass Software version 2.7.1 (ProteinSimple, Minneapolis, MN, USA) followed by a statistical analysis.

### 4.6. IL-8 Measurement

IL-8 was quantified using an AlphaLISA immunoassay kit (AL224C, Perkin-Elmer, Waltham, MA, USA) according to the manufacturer’s recommendations.

### 4.7. S. aureus Biofilm Formation

*S. aureus* (ATCC^®^ 35556™) was seeded at 0.1 million/mL within tryptic soy broth (Biomérieux^®^ TSB-F index 42614) enriched by glucose at 2% and incubated for 24 h at 37 °C. *Castanea sativa* extract was applied from the beginning to the end of the incubations. Biofilm formation was observed using crystal violet staining and a real-time, label-free technique based on impedance recording using an xCELLigence^®^ device (ACEA Biosciences, Santa Clara, CA, USA) as described [58].

### 4.8. Quantification of Enzymatic Activities Released by S. aureus

For lipase and hyaluronidase activities, various concentrations, from 0.165% (*v*/*v*) up to 5.5% and from 0.5% up to 5%, of the *Castanea sativa* extract was added in the culture broth with *S. aureus*. After 6 and 24 h, the number of bacteria was determined by densitometry at 600 nm, while supernatants were recovered by centrifugation of conditioned broth at 1000× *g* for 5 min and then used for enzymatic evaluation. Lipase activity was measured by recording the specific fluorescence of a synthetic substrate (Sigma #30058, St. Louis, MO, USA). Hyaluronidase activity was evaluated through the quantification of residual hyaluronic acid using a turbidimetric method [59]. For plasminogen activation, the *Castanea sativa* extract was incubated with *S. aureus*-conditioned broth at 0.015% to 0.15%. Then, plasminogen activation into plasmin was measured by recording the specific fluorescence of a protease substrate (Bodipy-Casein E6638, Sigma).

### 4.9. Cytokine Production by Human Keratinocytes

Keratinocytes (human cell line HaCaT) were seeded in a standard medium with fetal calf serum and incubated for 3 days at 37 °C. The growth medium was replaced by a standard medium containing the *Castanea sativa* extract at 0.55% or 1.65% and incubated for 24 h, and then cell layers were rinsed and infected by a defined dose of *S. aureus* for 2 h. Finally, the suspension of bacteria was exchanged for the standard medium containing the ingredient, and it was incubated once again for 24 h before evaluation of cell viability using the 3-(4,5-dimethylthiazol-2-yl)-2,5-diphenyltetrazolium bromide (MTT) test and cytokine release in supernatant medium using ELISA methods (IL-6 D6050 and IL-8 D8000C, R&D systems, Minneapolis, MN, USA).

### 4.10. IL-8 Released by Human Macrophages

Human macrophages (U937 cell line) were activated by phorbol myristate acetate for 48 h and then treated with the *Castanea sativa* extract for 24 h. The treatment medium was exchanged to a medium containing a defined quantity of heat-killed *S. aureus* bacteria, and it was incubated for 24 h at 37 °C. Cell viability was measured using the MTT test, while the release of IL-8 was measured on cell culture supernatant using the ELISA method (IL-8 D8000C, R&D Systems). The active ingredient was applied at 0.003% up to 0.03%.

### 4.11. Statistical Analysis

A statistical analysis was performed using Sigma Plot software with Student *t*-test, the Mann–Whitney test or One-way ANOVA test. A *p* value < 0.05 was considered significant.

## 5. Conclusions

In recent years, in vitro models reproducing some features of AD have been developed by challenging epidermis with either interleukin cocktails or *S. aureus* extracts or by silencing the expression of pivotal genes encoding epidermal barrier proteins. However, none of them reproduced all of the pathophysiological AD features. In this paper, we developed an AD-like model consisting of RHEs exposed to both a Th2 pro-inflammatory cytokine cocktail and *S. aureus*. This model mimics the impairments in the skin barrier observed in AD, at the physical, molecular and immune levels. Moreover, our results strongly suggest that *S. aureus* acquired a higher virulence potential when the epidermis was challenged with inflammatory cytokines, thus later contributing to disease exacerbation. The relevance of this model was confirmed using an extract of *Castanea sativa*, which significantly improved components of the physical, microbial and immune epidermal barriers.

## Figures and Tables

**Figure 1 ijms-23-12880-f001:**
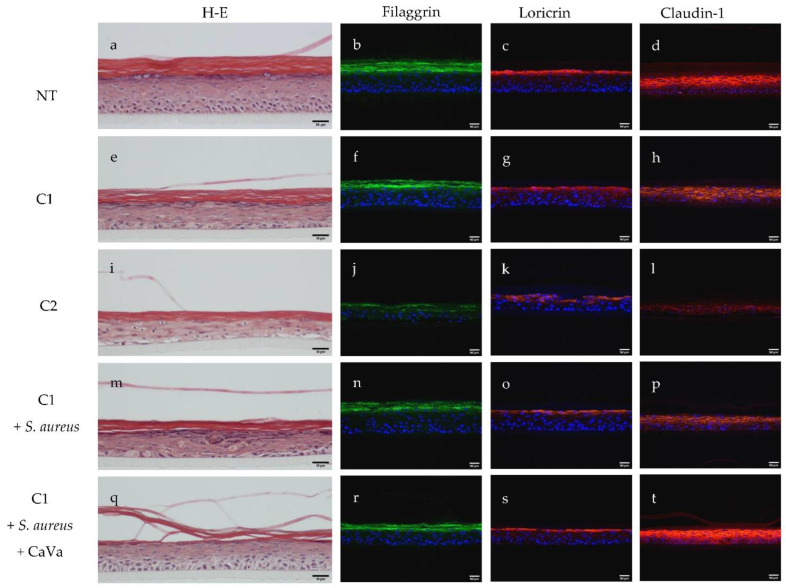
Effect of treatments of reconstructed human epidermis (RHE) with either a cocktail of inflammatory cytokines alone, both the cocktail and *S. aureus* or the cocktail with *S. aureus* and an extract of *Castanea sativa* leaves. RHEs were untreated (NT; (**a**–**d**)) or treated with TNFα and a cocktail of Th2-related cytokines (IL-4, IL-13 and Il-31) at 5 ng/mL (C1) or 20 ng/mL (C2) ((**e**–**h**) and (**i**–**l**), respectively). In some cases, *S. aureus* was topically applied on the RHEs treated with the lowest concentration of cytokines (**m**–**t**), without (**m**–**p**) or with (**q**–**t**) additional treatments with an extract of *Castanea sativa* leaves. Sections of RHEs were stained with hematoxylin–eosin (H-E; (**a**,**e**,**i**,**m**,**q**)) and immunodetected with antibodies directed to filaggrin ((**b**,**f**,**j**,**n**,**r**); green), loricrin ((**c**,**g**,**k**,**o**,**s**); red) and claudin-1 ((**d**,**h**,**l**,**p**,**t**); red). Nuclei were stained with 4′,6-diamidino-2-phenylindole (blue). Representative epifluorescence images are shown; scale bars = 20 µm.

**Figure 2 ijms-23-12880-f002:**
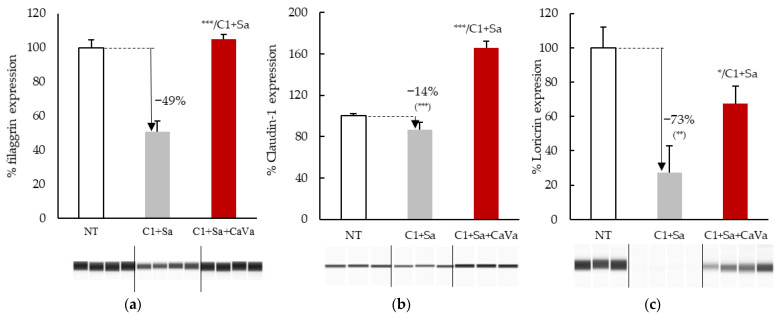
Modulation of the expressions of filaggrin (**a**), claudin-1 (**b**) and loricrin (**c**) in reconstructed human epidermis (RHE) treated with the inflammatory cocktail of cytokines and *S. aureus* (C1 + Sa), without and with an additional treatment with a *Castanea sativa* extract (CaVa). RHE total proteins were analyzed by immunoblotting (representative blots are shown), and the images were scanned to quantify the immunoreactive protein amounts. Mean values ± SD are shown, n = 3; Student *t*-test; *, *p* < 0.05; **, *p* < 0.01; ***, *p* < 0.001.

**Figure 3 ijms-23-12880-f003:**
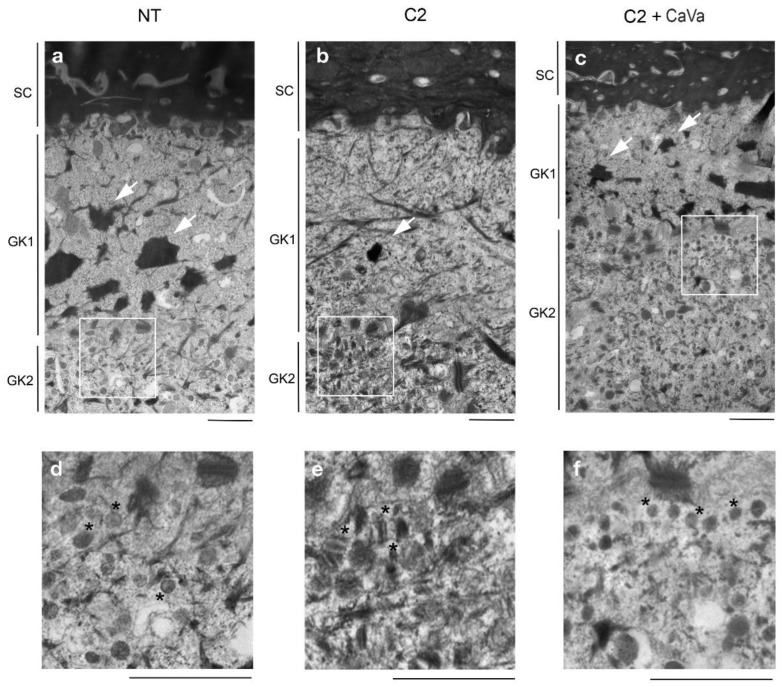
Ultrastructural analysis of reconstructed human epidermis (RHE). Non-treated (NT) RHEs (**a**) and RHEs treated with either the cocktail of cytokines at 20 ng/mL (C2) (**b**) or the cocktail and the extract of *Castanea sativa* (C2 + CaVa) (**c**) were analyzed using transmission electron microscopy. Scale bar = 1 µm. SC, *stratum corneum*; GK1, upper granular keratinocyte; GK2, second granular keratinocyte; *, lamellar bodies (**d**,**f**) or unidentified vesicles (**e**). Arrows show keratohyalin granules.

**Figure 4 ijms-23-12880-f004:**
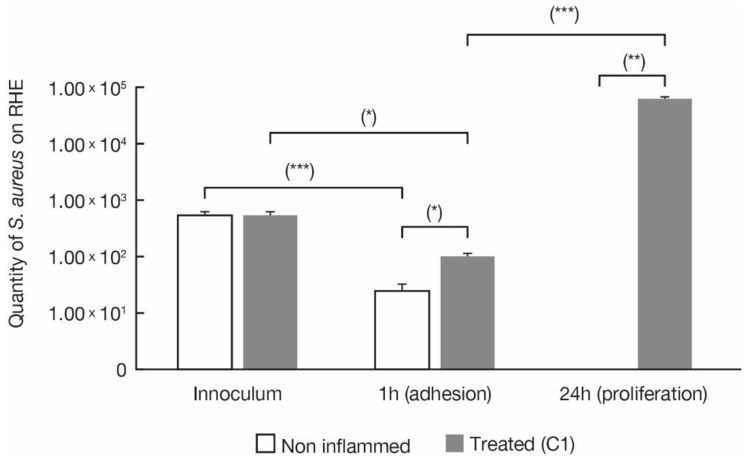
Promotion of *S. aureus* adhesion and proliferation by Th2 cytokines on reconstructed human epidermis (RHE) surface. *S. aureus* inoculum was topically applied on the surfaces of non-treated RHEs (open bars) and RHEs treated with 5 ng/mL of Th2 cytokines (C1; gray bars). The number of bacteria was measured 1 h and 24 h after inoculation (n = 3). Mean ± SEM, Student *t* test; *, *p* < 0.05; **, *p* < 0.01; ***, *p* < 0.001.

**Figure 5 ijms-23-12880-f005:**
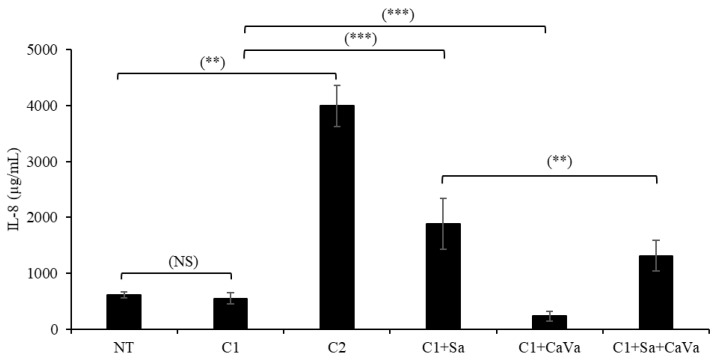
Induction of IL-8 release in RHEs treated with cytokines alone or with either *S. aureus* (Sa), *Castanea sativa* (CaVa) or both, as indicated (C1, 5 ng/mL; C2, 20 ng/mL). IL-8 secretion by keratinocytes was measured in RHE culture media by ELISA. n = 7–12, mean ± SD, Mann–Whitney. NS: not significant; **, *p* < 0.01; ***, *p* < 0.001.

**Figure 6 ijms-23-12880-f006:**
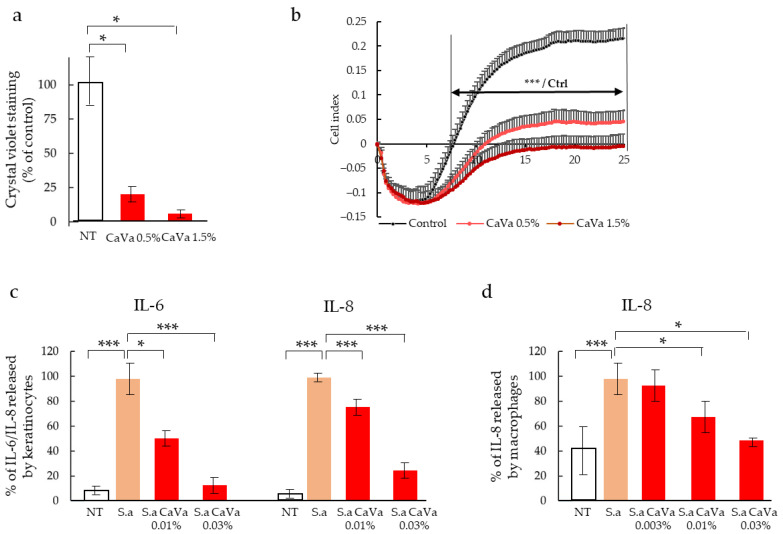
Decrease in *S. aureus* virulence induced by a *Castanea sativa* extract. (**a**,**b**) *S. aureus* was cultured within trypticase soy broth for 24 h at 37 °C without (NT) or with an extract of *Castanea Sativa* (CaVa) at the indicated concentration (n = 5). Biofilm formation was measured after crystal violet staining of adherent bacteria (**a**). Electric impedance reflecting instantaneous biofilm formation was recorded (**b**). Monolayer cultures of keratinocytes (**c**) and macrophages (**d**) were infected by *S. aureus* without (*S.a*) or with the plant extract, and release of IL-6 or IL-8 was quantified using ELISA (n = at least 3). Mann–Whitney test vs. control. Mean ± SD; *, *p* < 0.05; ***, *p* < 0.001. The highest amount of cytokines was arbitrarily fixed to 100.

## Data Availability

All data are included in the manuscript.

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
