# Peer review of "An Inflamed and Infected Reconstructed Human Epidermis to Study Atopic Dermatitis and Skin Care Ingredients"

_ijms, 2022, doi:10.3390/ijms232112880_

Round 1
Reviewer 1 Report
1) In general, atopic dermatitis models use a specific allergens such as avalbumin or house dust mite extract. Why didn't the authors include the most important allergen irritants in Th2 cytokine and S. aureus tissue environment?
2) As the inhibitory mechanism of CaVa in this model has not been studied, preliminary data or possible discussion on this should be mentioned.
3) It is known that the TSLP is very important in addition to the TH2 cytokine in the atopic dermatitis model. I think it would be good to discuss it.
Author Response
1/ We agree with this reviewer those specific allergens are used in in vivo mouse models. Here we are in vitro, in a simplified model where immune cells and allergen-specific IgE are not present, in order to analyze the effects on keratinocytes. However, to mimic an inflammatory state of the epidermis, and for reason of standardization of the model, we directly added soluble Th2 mediators in the culture medium.
2/ We add information regarding preliminary tests used to preselect the ingredient in the introduction line 97.
This active ingredient was preselected among a library of thousands of potential ingredients thanks to a preliminary anti-inflammatory property evaluation (quantification of IL6, IL8 by ELISA) on monolayer cultures of normal human keratinocytes stressed by poly I:C, TNF alpha and interferon gamma. Among anti-inflammatory hits, Castanea sativa extract was then selected for its capacity to inhibit IL6 and 8 productions in keratinocytes challenged by S. aureus ATCC35556, inhibit its lipase activity, and in parallel to stimulate filaggrin on differentiated keratinocytes.
We also add possible explanation of its action related to the phytochemical profile of the extract at the end of the discussion line 338.
Using UPLC-PDA we determined the flavonoid profile of the extract and particularly flavonoid glycosides and among them, a major compound Quercetin-3-O- glucuronide (miquelianin), but also Kampferol 3-O- glucuronide (astragalin). These are two compounds known as NFkB pathway modulators (Ha AT, et al., Int J Mol Sci 2022, 23, 433, https://doi.org/10.3390/ijms23010433; Mi XJ, et al. J Ethnopharmacol. 2022, 296:115523. doi: 10.1016/j.jep.2022.115523; Kim MS and Kim SH. Arch Pharm Res. 2011, 34, 2101–2107, https://doi.org/10.1007/s12272-011-1213-x; Soromou LW, et al. Biochem Biophys Res Commun. 2012, 419(2):256-61, doi: 10.1016/j.bbrc.2012.02.005). This may in part explain the effects of CaVa.
Ha, A.T.; Rahmawati, L.; You, L.; Hossain, M.A.; Kim, J.-H.; Cho, J.Y. Anti-Inflammatory, Antioxidant, Moisturizing, and Antimelanogenesis Effects of Quercetin 3-O-β-D-Glucuronide in Human Keratinocytes and Melanoma Cells via Activation of NF-κB and AP-1 Pathways. Int. J. Mol. Sci. 2022, 23, 433. https://doi.org/10.3390/ijms23010433
Mi XJ, Kim JK, Lee S, Moon SK, Kim YJ, Kim H. In vitro assessment of the anti-inflammatory and skin-moisturizing effects of Filipendula palmata (Pall.) Maxim. On human keratinocytes and identification of its bioactive phytochemicals. J Ethnopharmacol. 2022 Oct 5;296:115523. doi: 10.1016/j.jep.2022.115523. Epub 2022 Jul 7. PMID: 35809756.
Kim, MS., Kim, SH. Inhibitory effect of astragalin on expression of lipopolysaccharide-induced inflammatory mediators through NF-κB in macrophages. Arch. Pharm. Res. 34, 2101–2107 (2011). https://doi.org/10.1007/s12272-011-1213-x
Soromou LW, Chen N, Jiang L, Huo M, Wei M, Chu X, Millimouno FM, Feng H, Sidime Y, Deng X. Astragalin attenuates lipopolysaccharide-induced inflammatory responses by down-regulating NF-κB signaling pathway. Biochem Biophys Res Commun. 2012 Mar 9;419(2):256-61. doi: 10.1016/j.bbrc.2012.02.005. Epub 2012 Feb 10. PMID: 22342978.
3/ The TSLP is implicated in both Th17 inflammation and Th2 inflammation (giving rise to respectively psoriasis or AD phenotype for example). As we wanted to only focus on AD phenotype, we prefer to use strictly Th2 related mediators (Wang et al. 2021).
Wang SH, Zuo YG Thymic Stromal Lymphopoietin in Cutaneous Immune-Mediated Diseases. .Front Immunol. 2021 Jun 24;12:698522. doi: 10.3389/fimmu.2021.698522. eCollection 2021.
Reviewer 2 Report
The authors of the manuscript entitled "An inflamed and infected reconstructed human epidermis to study atopic dermatitis and skin care ingredients" described the results of a study on the use of a new 3D model of RHE aimed at mimicking the in vivo status of the known disease AD. The work is interesting, as are the results, however I feel that some parts need to be deepened and completed to make the work worthy of publication in a journal like IJMS.
Here are some suggestions:
-The authors in the Introduction should add some information regarding the choice of using the Castanea sativa extract: have you carried out any preliminary studies? Are there any data in the literature that support the use of this extract for AD? Could the authors provide some qualitative and quantitative indications relating to the composition of the extract?
What kind of extract is it? how is it applied on the RHE? Did the authors think about including it in a topical preparation for use in the treatment of AD?
-Line 105-106: the authors state "by hematoxylin-eosin staining (Fig. 1 a, e, i and m)", shouldn't the list in brackets also include "q"?
-Figure 2: why was the expression of the three proteins not quantified even after treatment with C2? why is there no comment in the text on the expression relating to the conc1 + Sa + CaVa treatment?
- Paragraph 2.3: should be paragraph 2.2 (therefore review all the numbering of the Results paragraphs). The authors here report the images relating to treatment with C2, unlike paragraph 2.1 which focused on treatments with C1. I would suggest that the authors standardize all results from the same concentration, or show both. Alternatively, enter a comment about the concentration not shown. In my opinion it would serve to make the results section more uniform.
-Line 318: Castanea sativa must be in italics
Author Response
-The authors in the Introduction should add some information regarding the choice of using the Castanea sativa extract: have you carried out any preliminary studies? Are there any data in the literature that support the use of this extract for AD? Could the authors provide some qualitative and quantitative indications relating to the composition of the extract?
We performed a preliminary screening of our library of thousands of ingredients regarding anti-inflammatory properties (quantification of IL6, IL8 by ELISA) on normal human cultures of keratinocytes stressed by a cocktail of poly I:C, TNF alpha and interferon gamma.
Among anti-inflammatory hits, castanea sativa extract was then selected for its capacity to inhibit IL6 and 8 productions in keratinocytes challenged by S. aureus ATCC35556 and its lipase activity, in parallel to stimulate filaggrin on differentiated keratinocytes.
Precisions were added in the text line 97.
Using UPLC-PDA we determined the flavonoid profile of the extract and particularly flavonoid glycosides and among them, a major compound Quercetin-3-O- glucuronide (miquelianin), but also Kampferol 3-O- glucuronide (astragalin). These are two compounds known as NFkB pathway modulators (Ha AT, et al., Int J Mol Sci 2022, 23, 433, https://doi.org/10.3390/ijms23010433; Mi XJ, et al. J Ethnopharmacol. 2022, 296:115523. doi: 10.1016/j.jep.2022.115523; Kim MS and Kim SH. Arch Pharm Res. 2011, 34, 2101–2107, https://doi.org/10.1007/s12272-011-1213-x; Soromou LW, et al. Biochem Biophys Res Commun. 2012, 419(2):256-61, doi: 10.1016/j.bbrc.2012.02.005). This may in part explain the effects of CaVa.
This was added at the end of the discussion section of the revised manuscript, line 338.
Since the potential activity to help resolving AD of the Castanea sativa extract was demonstrated in this study, a patent was applied (in December 2021).
What kind of extract is it? how is it applied on the RHE? Did the authors think about including it in a topical preparation for use in the treatment of AD?
The Castanea extract is an extract of organic certified leaves of chestnut trees characterized by the flavonol glycosides Miquelianin and Astragalin.
The Castanea extract was added to the culture medium of RHE to avoid damaging them by every two days application. The systemic way of treatment was added line 365.
However clinical testing was performed in a cosmetically acceptable formulation topically applied. The base cream in which we incorporated 2% Castanea extract was an emulsion inspired by an established market reference product for very dry skin.
-Line 105-106/114: the authors state "by hematoxylin-eosin staining (Fig. 1 a, e, i and m)", shouldn't the list in brackets also include "q"?
You are right “q” has been added
-Figure 2: why was the expression of the three proteins not quantified even after treatment with C2? why is there no comment in the text on the expression relating to the conc1 + Sa + CaVa treatment?
To set up the method, we tested first filaggrin by western blot for the conditions C1+SA and C2, with or without CaVa extract. As the decrease observed for C2 and C1+SA were equivalent as the restoration by the CaVa extract, due to lack of samples to be treated and analyzed for all conditions and methods in the final experimentation, for western blotting, we unfortunately only run the 3 proteins for the conditions untreated, C1+SA and C1+SA+CaVa.
The effect of CaVa on C1+Sa treated keratinocytes has been described in the text, at the end of the first paragraph of section 2.5 A Castanea Sativa extract reverses the AD-like phenotype of Th2 cytokines- and S. aureus-treated RHEs: “The induced down-regulation of filaggrin, loricrin and claudin-1 expression was reversed as shown using indirect immunofluorescence (Fig. 1 r-t) and western blotting (Fig. 2) analysis)”. This reviewer probably missed the sentence.
- Paragraph 2.3: should be paragraph 2.2 (therefore review all the numbering of the Results paragraphs). The authors here report the images relating to treatment with C2, unlike paragraph 2.1 which focused on treatments with C1.
I would suggest that the authors standardize all results from the same concentration, or show both. Alternatively, enter a comment about the concentration not shown. In my opinion it would serve to make the results section more uniform.
Unfortunately, due to lack of samples, we preferred to evaluate in TEM the most deleterious condition previously observed in histology and immunostaining (ie C2).
-Line 318/344: Castanea sativa must be in italics
You are right, this was modified